# Contribution of a Novel *B3GLCT* Variant to Peters Plus Syndrome Discovered by a Combination of Next-Generation Sequencing and Automated Text Mining

**DOI:** 10.3390/ijms20236006

**Published:** 2019-11-28

**Authors:** Justyna Totoń-Żurańska, Przemysław Kapusta, Magda Rybak-Krzyszkowska, Katarzyna Lorenc, Julita Machlowska, Anna Skalniak, Erita Filipek, Dorota Pawlik, Paweł P. Wołkow

**Affiliations:** 1Center for Medical Genomics–OMICRON, Jagiellonian University Medical College, ul. Kopernika 7c, 31–034 Krakow, Poland; jzuranska@gmail.com (J.T.-Ż.); przemyslaw.kapusta@uj.edu.pl (P.K.); julita.machlowska@gmail.com (J.M.); 2Department of Obstetrics and Perinatology, The University Hospital in Krakow, ul. Kopernika 23, 31-501 Krakow, Poland; rybaczka@interia.pl; 3Department of Neonatology, Jagiellonian University Medical College, ul. Kopernika 23, 31-501 Krakow, Poland; lorenc.kasia@poczta.onet.pl (K.L.); dorota.pawlik@uj.edu.pl (D.P.); 4Department of Endocrinology, Jagiellonian University Medical College, ul. Kopernika 17, 31–501 Krakow, Poland; anna.skalniak@uj.edu.pl; 5Clinic and Department of Pediatric Ophthalmology, School of Medicine in Katowice, Medical University of Silesia in Katowice, ul. Ceglana 35, 40-514 Katowice, Poland; erita.filipek@gmail.com

**Keywords:** clinical genetics, diagnosis, anterior segment disease, ophthalmology

## Abstract

Anterior segment dysgenesis (ASD) encompasses a spectrum of ocular disorders affecting the structures of the anterior eye chamber. Mutations in several genes, involved in eye development, are implicated in this disorder. ASD is often accompanied by diverse multisystemic symptoms and another genetic cause, such as variants in genes encoding collagen type IV. Thus, a wide spectrum of phenotypes and underlying genetic diversity make fast and proper diagnosis challenging. Here, we used AMELIE, an automatic text mining tool that enriches data with the most up-to-date information from literature, and wANNOVAR, which is based on well-documented databases and incorporates variant filtering strategy to identify genetic variants responsible for severely-manifested ASD in a newborn child. This strategy, applied to trio sequencing data in compliance with ACMG 2015 guidelines, helped us find two compound heterozygous variants of the *B3GLCT* gene, of which c.660+1G>A (rs80338851) was previously associated with the phenotype of Peters plus syndrome (PPS), while the second, NM_194318.3:c.755delC (p.T252fs), in exon 9 of the same gene was noted for the first time. PPS, a very rare subtype of ASD, is a glycosylation disorder, where the dysfunctional *B3GLCT* gene product, O-fucose-specific β-1,3-glucosyltransferase, is ineffective in providing a noncanonical quality control system for proper protein folding in cells. Our study expands the mutation spectrum of the *B3GLCT* gene related to PPS. We suggest that the implementation of automatic text mining tools in combination with careful variant filtering could help translate sequencing results into diagnosis, thus, considerably accelerating the diagnostic process and, thereby, improving patient management.

## 1. Introduction

In recent years, next-generation sequencing technology (NGS) and access to databases containing sequencing data from patients with eye disorders have accelerated research in ophthalmic molecular genetics. However, still the most difficult and time-consuming part of the work is data analysis and clinical interpretation of reported variants, especially in cases with blended phenotype, when the decision on major clinical feature, crucial for diagnostic process is difficult to make. Additionally, due to massive sequencing data generation and relatively restricted curation, available databases, within reliable information contain false variant pathogenicity attributions. Thus, rigorous variant interpretation, as recommended by the American College of Medical Genetics and Genomics (ACMG) [1], followed by extensive review of the existing literature could improve diagnostic accuracy.

Anterior segment dysgenesis (ASD) encompasses a spectrum of ocular disorders affecting the structures of the anterior eye chamber, including the iris, lens, cornea, and the anterior chamber angle. Clinical manifestations include corneal opacities, cataracts, and iridocorneal adhesions. ASD poses a high risk of the patient developing early onset and aggressive glaucoma resulting from dysregulation of aqueous humor flow, high intraocular pressure (IOP), and death of retinal ganglion cells [2]. Disorders related to such a phenotype include Axenfeld Rieger syndrome (ARS), isolated Peters Anomaly, Peters plus syndrome (PPS), primary congenital glaucoma, congenital hereditary endothelial dystrophy, and iridogoniodysgenesis anomaly [3], all of which are registered as rare by Orphanet. The genetic background of ASD is only partially known and may be related to the disruption of many genes, e.g., *CYP1B1*, *BMP4*, *FOXC1*, *PAX6*, *FOXE3*, *NDP*, *SLC4A11*, *HCCS*, *PITX2*, *PITX3*, *LMX1B,* and *PXDN* [4,5,6,7,8], several of which can modulate tyrosinase activity, postulated as an important modulator of iridocorneal angle malformations [6]. Importantly, diseases with multisystemic manifestations can also present with ASD, e.g., syndromes resulting from mutations in genes encoding collagen type IV: *COL4A1* and *COL4A2* [2,9,10], often manifest with neurological disorders, what may hinder finding the relevant genetic cause of the disease. Diseases from the ASD spectrum may be dominant (ARS, aniridia) or recessive (primary congenital glaucoma, PPS), and due to the diverse genetic background, several phenotypes may present with either mode of inheritance [11].

Due to the heterogeneous clinical manifestation of the syndrome, reflecting the complex genetic background, classification, and proper diagnosis of ASD is still challenging. Here, we report a strategy for NGS data analysis in order to find the genetic cause of a rare combination of symptoms in a pediatric patient with a set of ocular disorders, cleft lip and high-arched palate, using data from DNA sequencing of coding regions of 4813 genes in both parents and the affected child.

We demonstrate a variant filtering strategy combined with information on the observed phenotype coded in Human Phenotype Ontology (HPO) terms and information enrichment by literature mining with the AMELIE (Automatic MEndelian LIterature Evaluation) [12] tool and by genomic variant annotation and prioritization with wANNOVAR [13,14,15] to help draw biological insights from sequencing data and increase the chances of faster and accurate diagnosis. Literature mining tool prioritizes variants according to the current knowledge from published data and enables critical verification of results through linking ranked genes important for the phenotype with relevant publication, what accelerates the process and may reduce false positive findings. In our opinion, this analytical strategy should be applicable to other clinical scenarios with complex genetic background and heterogeneous clinical manifestations also.

## 2. Results

### 2.1. Case History and Clinical Findings

#### 2.1.1. Obstetric Information

The 30-year-old pregnant patient was referred for the first-trimester testing due to three previous miscarriages and advanced maternal age. The parents had one healthy child. Screening involved testing for trisomies 21, 18, and 13, ultrasonography, measurement of maternal serum free β-human chorionic gonadotropin (β-hCG), and pregnancy-associated plasma protein-A (PAPP-A) levels, according to the Fetal Medicine Foundation recommendations. Both biochemical parameters were measured on the DELFIA Xpress analyzer (Perkin Elmer). The mother denied medication history or infection during pregnancy. Moreover, no familial predisposition to any disease was identified. Routine fetal scanning at 13 weeks of gestation showed a cleft lip and palate and increased nuchal translucency (Figure 1). Nuchal translucency was assessed as 3 mm and was estimated as being over the 95th percentile for the given crown-rump length (CRL = 67.7 mm). No other anomalies were detected for the fetus. Free β-hCG was 1.472 MoM (Multiple of *Median*) and PAPP-A, 0.634 MoM. The first trimester combined with prenatal screening yielded a risk of 1:149 for trisomy 21, 1:7780 for trisomy 18, and 1:7447 for trisomy 13.

Having detected a fetal defect and correlated it with an indirect risk group, an invasive diagnostic procedure was suggested to determine the fetal karyotype. Amniotic fluid was aspirated during amniocentesis on week 15, day 4 of pregnancy. Following amniocyte culture, a karyotype of a normal male (46, XY) was obtained at a resolution of 550 bands. The couple was then referred for genetic counseling for further etiological investigation.

The second ultrasonographic examination performed at the 20th week of gestation, as per the recommendations of the Polish Society of Gynecologists and Obstetricians, confirmed the presence of bilateral cleft lip and palate.

The patient (mother) was hospitalized with intrauterine growth restriction in the 28th week of gestation. In view of abnormal flow in the umbilical artery and the middle cerebral artery, as well as deteriorating fetal well-being, it was decided to terminate the pregnancy and perform a cesarean section.

#### 2.1.2. Initial Postnatal Examination

Physical examination of the fetus revealed bilateral cleft alveolar processes, cleft lip, and high-arched palate. Rhizomelic proximal extremities and brachydactyly were noted. Microphthalmia and bilateral corneal opacities were observed during ophthalmic examination.

#### 2.1.3. Ophthalmic Evaluation

Visual acuity in both eyes was determined as dubious light perception. Examination under general anesthesia resulted in the following findings (Figure 2).

Right eye (RE): Cornea dimensions 8 x 8.5 mm; IOP 10 mmHg; central corneal leucoma, subsequent to ulceration with ingrowth of blood vessels. Translucent peripheral cornea revealed flattened anterior eye chamber, medium-width iris with persistent pupillary membrane, and lens adhesion to the posterior corneal surface. A visible fragment of the lens was translucent with pink reflection from the eye fundus. Further details were hard to evaluate.

Left eye (LE): Cornea dimensions 8 × 8.5 mm; IOP 12 mmHg, partial central leucoma. The peripheral part of the cornea was translucent with a persistent pupillary membrane. The pupillary dilatory response was normal and some peripheral blood vessels of the retina could be visualized.

The anatomical axial length of the eyeballs in ultrasonographic (USG) evaluation was 18 mm (RE) and 17.5 mm (LE) in projection AB (Ultrascan B, Alcon, USA). Diagnostic USG of the posterior part of the eyes revealed no pathology in RE and a small floater above an optic nerve disk in LE.

The patient remained under ophthalmic supervision. At the age of four months, central thinning of the corneal leucoma of the RE, threatening perforation and secondary glaucoma were determined in the RE. Corneal thickness in the RE was 826 µm in the periphery but only 158–380 µm in the center while being 855 µm in the entire LE. IOP was 37 mmHg in the right and 17 mmHg in the LE. Amniotic membrane transplantation to protect the cornea and cyclocryotherapy to lower IOP were performed in the RE. Topical IOP lowering medications were started and the patient was followed for local status and IOP. In spite of ongoing pharmacological treatment, another round of amniotic membrane transplantation and cyclocryotherapy had to be performed at the age of six months. Stabilization of the IOP was achieved at the level of 26 mmHg in the right and 18 mmHg in the LE. The patient remains under ophthalmic supervision with permanent pharmacological medications to prevent glaucoma of the RE.

### 2.2. High-Throughput Sequencing Results

On average, 9.8 million reads were sequenced per sample, and approximately 8.9 million reads (91.1%) were mapped on the reference using our custom pipeline. A high depth of coverage (at least 20×) was maintained for 86.9% of the targeted region and approximately 95% of the target region was covered at least 10×. In total, we detected 32,222 variants: 26,230 single nucleotide polymorphisms (SNPs), 2536 insertions, and 3456 deletions.

Following the GEMINI trio analysis, we identified 265 variants in total, of which 36 were de novo variants, 200 autosomal recessive variants, and 16 pairs of compound heterozygotes (32 variants, of which three were de novo) in the proband. In terms of localization, 38 variants were annotated as exonic and 227 as intronic changes. All discovered variants were concatenated into one VCF file for further analysis (Appendix A). Using in silico prediction algorithms such as SIFT, PROVEAN, FATHMM-XF, and MutationTaster, we were able to select several candidates with possible damaging impact (Table 1). For genomic variant prioritization, we uploaded the generated VCF file (Appendix A) into AMELIE and wANNOVAR with the following HPO identifiers: Corneal opacity (HP:0007957), abnormality of the anterior chamber (HP:0000593), anterior chamber synechiae (HP:0007833), and cleft lip/palate (HP:0000202). Based on AMELIE and wANNOVAR results (Table 2), we discovered that variants in both alleles of *B3GLCT* (alias *B3GALTL*) that were segregated from an unaffected father and unaffected mother, were strong candidates to be potentially responsible for the observed pathogenicity, which, most likely, was PPS (Appendix A).

The first variant, inherited from the mother, was a substitution, NM_194318.3:c.660+1G>A (rs80338851), reported as pathogenic allele (*B3GALTL*, OMIM:610308:0001, RCV000001326.4) associated with PPS. Moreover, using Varsome, the discovered variant was classified according to ACMG2015 as pathogenic, with identified criteria to be PVS1, PP3, PP5.

The second variant, inherited from the father, was a deletion, NM_194318.3:c.755delC (p.T252fs), which has not been previously reported (according to LOVD, and is not present in the gnomAD database). In Varsome, the discovered variant was classified according to ACMG2015 as likely pathogenic, with identified criteria to be PVS1 and PM2. Moreover, this mutation had a MutationTaster score of 1 (prediction: Disease causing) and changed the reading frame downstream of the amino acid (AA) at position 252. As a result, the mutation terminated translation at AA position 264, whereas a wild-type protein is 498 AA long. Both mutations were manually reviewed in an integrative genomic viewer (IGV).

### 2.3. Sanger Sequencing Results

For final confirmation of the potentially causative variant in *B3GLCT,* we performed Sanger sequencing of exons 8 and 9 in the patient and both parents. Both mutations were present in the patient and the inheritance pattern was confirmed. Sanger sequencing electropherograms are presented in Figure 3.

## 3. Discussion

As reported, 6%–8% of children are born with developmental disorders, with many of them caused by genetic changes in a single gene [16]. Despite substantial progress in the NGS data analysis field, finding the causal variant is still a time-consuming process, with the diagnostic yield falling between 25% and 30% for exome sequencing data [17,18,19], especially due to difficulties in the interpretation of the functions of rare variants. In the current study, we used AMELIE and wANNOVAR on NGS data to prioritize genetic variants based on the putative contribution to the clinical phenotype of a pediatric patient suffering from anterior chamber disorder involving corneal opacity, persistent pupillary membrane, thinning of the posterior cornea and lenticulo-corneal adhesions with secondary glaucoma and accompanying cleft lip and palate.

Independent of the sequencing scale, NGS creates considerable data burden Thus, careful variant filtering and prioritization, according to the information available from sequencing databases, in compliance with ACMG standards are required. Additionally, differences in NGS data analysis between laboratories could generate discrepancies in the diagnosis, thus filtering schemes should be clearly reported. Moreover, clinical interpretation of sequencing data requires matching information on patients’ genotypes and phenotypes and is usually complemented by the integration of information from databases such as OMIM or ClinVar, which are manually constructed and curated. The search for causal variants should be supported by published evidence if it exists, and this requires manual querying of relevant literature databases. This step is the most time-consuming in data processing and has a major impact on timely diagnosis and minimizing false negative and false positive findings. Web-based literature search engines are employed to assist manual curation of newly published information. AMELIE is a method for ranking candidate causal genes related to the phenotype, extracted from primary literature using supervised machine learning methods. The inclusion of HPO-encoded phenotype information facilitates communication with clinical specialists. In our opinion, it combines the convenience of other HPO-based variant ranking tools such as Exomiser [20] with enrichment of the latest literatures. wANNOVAR is a web server which provides functional annotation of genetic variants, reporting their conservation levels, calculating their predicted functional importance scores, retrieving allele frequencies in public databases, and implementing a protocol to identify a subset of potentially deleterious variants/genes.

The combination of these two strategies applied to trio sequencing data led us to the identification of compound heterozygous variants in the *B3GLCT* gene, which has been previously associated with the PPS phenotype, observed in our patient. Using only a VCF file, annotated by GEMINI with several pathogenicity prediction tools (SIFT, Provean, FATHMM-XF, MutationTaster), resulted in conflicting predictions for variants in some cases. Although the list of possible causative variants included two variants of *B3GLCT* as compound heterozygous variants, it was not fully informative as the annotation missed the prediction for the deletion NM_194318:c.755delC. Both AMELIE and wANNOVAR ranked variants in genes related to HPO-encoded phenotypes. As a result, *B3GLCT* got the highest score with both tools. Moreover, in contrast to lower ranked genes, the disease annotated to *B3GLCT*—PPS-matched our patient’s phenotype perfectly.

PPS, a subtype of ASD, inherited in a recessive manner, is a very rare disease with unknown incidence or prevalence. As reported in 2016 by Jaak Jaeken et al., the worldwide number of patients diagnosed with PPS due to *B3GLCT* mutation was 49 [21]. Common features of PPS involve anterior chamber dysgenesis symptoms such as central corneal clouding, cataract, thinning of the posterior surface of the cornea, iris hypoplasia, and iridocorneal adhesions. In addition to ocular defects, PPS is characterized by short stature, dysmorphic facial features, and developmental delay [21]. In terms of classification, PPS belongs to genetically heterogeneous congenital disorders of glycosylation, grouping rare congenital, neurometabolic, and malformation syndromes [22,23]. It is caused by changes in *B3GLCT,* encoding an *O*-fucose-specific β-1,3-glucosyltransferase (beta-1,3-glucosyltransferase), responsible for attachment of glucose to *O*-linked fucose (*O*-fucose), which is previously added by protein *O*-fucosyltransferase 2 (*POFUT2*) to thrombospondin type 1 repeats (TSRs), present in many proteins [24]. Localized at the endoplasmic reticulum (ER) [25], beta-1,3-glucosyltransferase modifies only properly folded TSRs and promotes secretion of ER proteins stabilized by glycosylation. Together with *POFUT2*, *B3GLCT* provides a noncanonical quality control system of proper protein folding in cells [26]. A GWAS study on more than 17,100 patients, with advanced age-related macular degeneration (AMD) and over 60,000 controls, showed a significant association between AMD and loci in *B3GLCT* and *ADAMTS9* [27].

On that date, there were 22 unique variants of the *B3GLCT* gene reported in 10 individuals in the LOVD database. Variant c.660+1G>A (rs80338851), located at the donor splice site (5′ss) of exon 8 observed in our patient, is one of the most common among variants identified in PPS patients [28]. It has a frequency of 0.01% in the gnomAD database for European (non-Finnish) population and accounts for 69% of all reported pathogenic alleles [21]. The results of Oberstein et al. showed that c.660+1G>A alters the acceptor site of exon 8, leading to the skipping of this exon and the introduction of a premature termination codon (PTC) at position +10 within exon 9. According to the position rule for PTC, c.660+1G>A results in a nonsense mRNA that elicits nonsense-mediated mRNA decay (NMD) [29]. NMD is the surveillance pathway protecting cells from the action of truncated proteins which could be translated from transcripts bearing a PTC [30]. In mammalian cells, it is the process of degradation of the mRNA, depending on the interaction between translation termination, complex multiprotein assemblies, which takes place when PTC is present at least 50 nucleotides upstream of the last exon junction. mRNA degradation due to NMD is involved in many neurological and developmental disorders [31].

The second of the detected variants, c.755delC (p.Thr252fs), was not present in any publicly available database and is the first one reported in exon 9. Pathogenicity prediction algorithms assigned this variant as disrupting. Similar to c.660+1G>A, c.755delC introduces a PTC within exon 9, 26 nucleotides from the splice site, which also should result in NMD.

Our study demonstrates the important role and possible diagnostic utility of NGS combined with AMELIE and wANNOVAR.

Such an approach, on the one hand, brings data on genes and variants with well-described roles in disease development based on curated, authoritative databases such as OMIM (wANNOVAR) and, on the other hand, updates the analysis with current information from published manuscripts. Such a procedure helps reduce the number of false positive findings not related to a phenotype which are over-abundant in the VCF file annotated with pathogenicity tools.

## 4. Methods

### 4.1. Clinical Assessment

The patient was recruited from the Clinic of Neonatology at the Jagiellonian University Medical College, Krakow, Poland and the Ophthalmology Clinic and Department of Ophthalmology, Medical University of Silesia in Katowice, Poland. Written informed consent was obtained from the study participants and informed parental consent was obtained on behalf of the child. Developmental and dysmorphology assessments were conducted by a clinical geneticist. This research adhered to the tenets of the Declaration of Helsinki and was performed upon approval of the protocol by the Jagiellonian University Ethics Committee No. 122.6120.12.2015 (29 January 2015).

### 4.2. Ophthalmic Evaluation

The patient was admitted to the outpatient clinic of the Department of Ophthalmology at the Silesian Medical University in Katowice at the age of seven weeks and then, subsequently, at four and six months.

### 4.3. DNA Extraction, Library Preparation, and NGS

DNA was extracted from the peripheral blood of the affected child and both his healthy parents with the Maxwell 16 Blood DNA purification kit (Promega, Madison, WI, USA) on the Maxwell 16 device (Promega). Sequencing libraries were prepared with the TruSight one sequencing panel kit (Illumina, San Diego, CA, USA) according to the manufacturer’s protocol. In brief, 50 ng of DNA was fragmented and adaptor-tagged in an enzymatic reaction. Genomic libraries were enriched for the coding regions of the 4813 genes by two cycles of hybridization with biotinylated probes, followed by capture on streptavidin beads; 12.5 pM libraries were sequenced on the MiSeq sequencer (Illumina) using v3 chemistry reagents (2 × 150 bp reads).

### 4.4. Data Analysis

Raw reads were processed with the Illumina software, generating base calls and corresponding base-call quality scores. These data were then processed through our custom pipeline that uses open-source programs (Figure 4). Briefly, generated fastq files were fed to the FastQC software (version 0.11.5) to provide quality control checks on sequenced data (Andrews et al., 2010). Reads were aligned to the human reference genome GRCh37 (hg19) using the BWA-MEM algorithm from the Burrows Wheeler Aligner (BWA, version 0.7.5) [32]. Unmapped, low mapping quality score and duplicated reads were filtered out with SAMtools (version 0.1.19) [33]. GATK (version 3.7) base quality recalibration was applied across all samples simultaneously using variant quality score recalibration (VQSR) according to the GATK best practices recommendations [34,35]. Filtered variants were concatenated into one record (VCF file) and then, using GEnome MINIng (GEMINI, version 0.18.3) [36], the discovered variants were annotated with SnpEff (version 4.2) [37], and loaded into the SQLite database for parents-child trio analysis. Using the function for trio analysis implemented in the GEMINI software, we identified three groups of variants in the proband patient: (1) De novo, (2) autosomal recessive, and (3) compound heterozygotes. All variants were concatenated into one VCF file and used to predict in silico the effects of amino acid substitutions and indels using SIFT [38], Provean [39], FATHMM_XF [40], and MutationTaster [41]. Simultaneously, the generated VCF file was uploaded into two web tools, Automatic Mendelian Literature Evaluation (AMELIE) [12] and wANNOVAR [15], which connect the ANNOVAR [13] annotation pipeline with the Phenolyzer [42] gene prioritization pipeline. Both softwares were used with standard settings and patient phenotype identifiers as HPO ID. Finally, predicted causative variants were classified according to ACMG2015 guidelines, to verify potential pathogenicity using Varsome [43] online tools.

### 4.5. Sanger Sequencing

For final confirmation of the most plausible causal variants, exons 8 and 9 of *B3GLCT* (RefSeq NM_194318) were analyzed by Sanger sequencing on the ABI3500 sequencer (Applied Biosystems, ThermoFisher Scientific, Waltham, MA, USA). The obtained sequences were aligned to the reference NC_000013 with the SeqScape software (LifeTechnologies, ThermoFisher Scientific). After identification of an exon-shortening event, the overlapping sequence resulting from the heterozygous deletion was analyzed manually using SnapGene (GSL Biotech, Chicago, IL, USA).

## Figures and Tables

**Figure 1 ijms-20-06006-f001:**
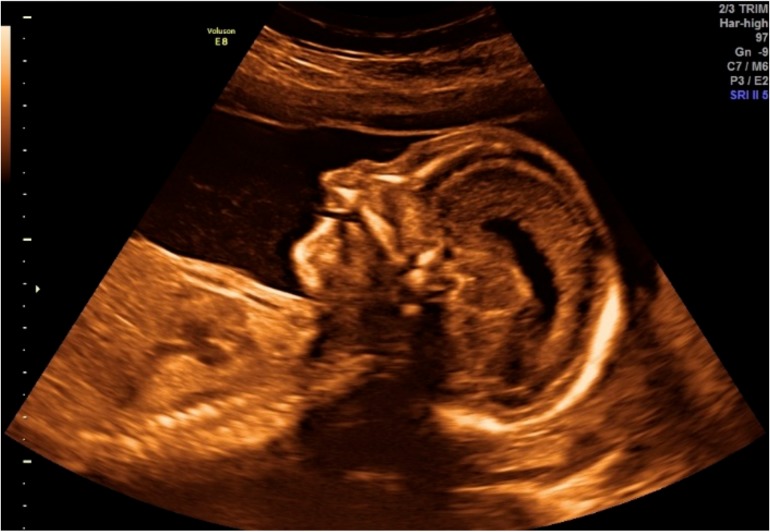
Obstetric sonogram of the fetus at 13 weeks of gestation.

**Figure 2 ijms-20-06006-f002:**
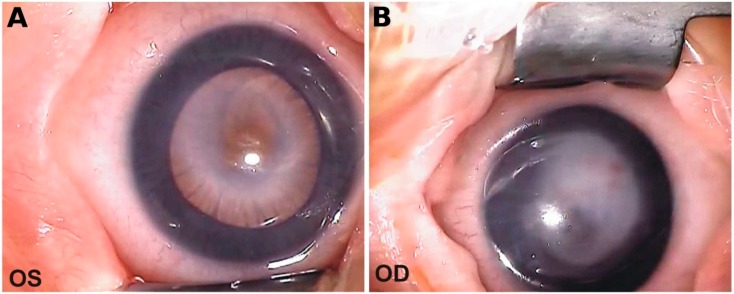
Anterior segment images of left (**A**) and right (**B**) eyes. Scale bars represent 2 mm.

**Figure 3 ijms-20-06006-f003:**
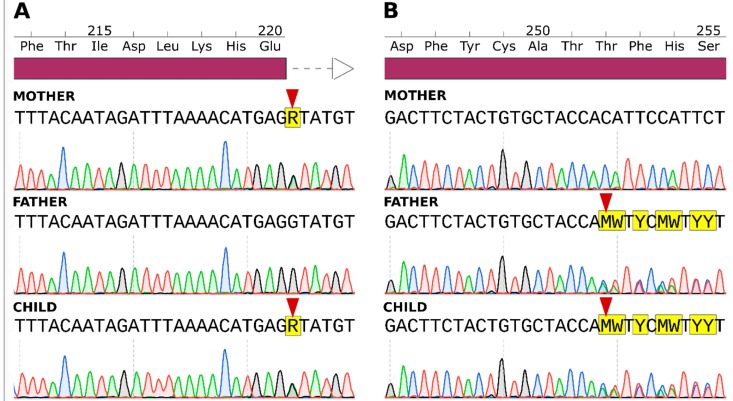
Sanger sequencing electropherograms of *B3GLCT* exon 8 (**A**) and exon 9 (**B**). Points of mutations are indicated with red arrows (violet bar indicates exonic region, white arrow intronic region).

**Figure 4 ijms-20-06006-f004:**
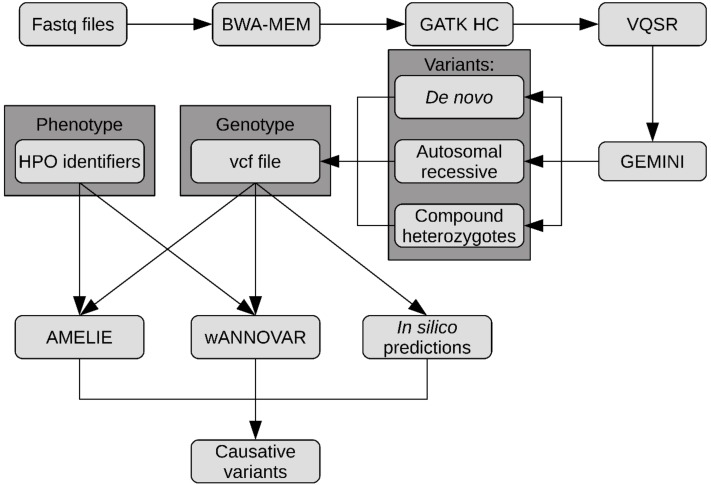
Flow chart describing the steps in the data analysis pipeline.

**Table 1 ijms-20-06006-t001:** Mutations discovered in the proband with possible damaging impact according to in silico analysis. Variants were sorted according to genomic coordinates.

Gene	Mutation	Amino Acid Change	Zygosity	Segregation Analysis	gnomAD European MAF	SIFT	PROVEAN	FATHMM-XF	Mutation Taster	dbSNP
*LRP8*	NM_004631:c.73_74delCA	p.Gln25fs*10	Compound Heterozygous	De novo	3.55 × 10^−4^	–	–	–	Disease causing	rs1491461533
	NM_004631:c.71delT	p.Leu24fs*50		De novo	1.79 × 10^−4^	–	–	–	Disease causing	rs761955852
*TTN*	NM_133437:c.71141G>A	p.Arg23714His	Compound Heterozygous	Maternal	2.88 × 10^−3^	Tolerated	Neutral	Benign	Polymorphism	rs200650668
	NM_133437:c.25190G>T	p.Ser8397Ile		Paternal	8.52 × 10^−4^	–	Deleterious	–	Disease causing	rs200335120
*DMXL1*	NM_005509:c.7403_7405delATG	p.Asp2468del	Heterozygous	De novo	9.30 × 10^−5^	–	Deleterious	–	Disease causing	rs200335120
*LAMA4*	NM_001105206:c.4665+7T>C	–	Compound Heterozygous	Maternal	1.80 × 10^−5^	–	–	Pathogenic	Disease causing	rs751477013
	NM_001105206:c.3239G>A	p.Arg1080Gln		Paternal	1.39 × 10^−2^	Damaging	Neutral	Pathogenic	Disease causing	rs41289902
*DEAF1*	NM_021008.3:c.69_98del30	p.Ala24_Ala33del	Heterozygous	De novo	1.06 × 10^−3^	–	–	–	Disease causing	rs766934551
*GYS2*	NM_021957:c.421G>A	p.Gly141Ser	Homozygous	Maternal,	2.12 × 10^−3^	Tolerated	Deleterious	Pathogenic	Disease causing	rs149533049
				Paternal						
*B3GALTL*	NM_194318:c.660+1G>A	–	Compound Heterozygous	Paternal	1.09 × 10^−3^	–	–	Pathogenic (high conf.)	Disease causing	rs80338851
	NM_194318:c.755delC	p.Thr252fs*13		Maternal	–	–	–	–	Disease causing	–
*RYR3*	NM_001036:c.9355G>A	p.Glu3119Lys	Compound Heterozygous	Maternal	3.06 × 10^−3^	Tolerated	Deleterious	Pathogenic	Disease causing	rs200830195
	NM_001036:c.14110G>A	p.Glu4704Lys		Paternal	1.08 × 10^−2^	Tolerated	Neutral	Pathogenic	Disease causing	rs182257230
*KIR2DL3*	NM_015868:c.44T>G	p.Leu15Trp	Heterozygous	De novo	1.07 × 10^−3^	Damaging	Deleterious	Benign	Polymorphism	rs149183658
	NM_015868:c.70+3G>A	–	Heterozygous	De novo	1.34 × 10^−5^	–	–	Pathogenic	–	rs776793416
*CELSR1*	NM_014246:c.8282C>T	p.Ser2761Leu	Compound Heterozygous	Maternal	3.24 × 10^−3^	Damaging	Deleterious	Pathogenic	Disease causing	rs144039991
	NM_014246:c.7313G>A	p.Arg2438Gln		Paternal	6.34 × 10−^4^	Tolerated	Neutral	Benign	Polymorphism	rs199688538

**Table 2 ijms-20-06006-t002:** Results from variant mining with AMELIE and wANNOVAR software. We have showed the top five genes from each software. Variants were sorted according to AMELIE best score.

Gene	Best AMELIE Score	AMELIE Rank	Phenolyzer Score	Phenolyzer Rank
*B3GLCT*	99.3	1	0.780	1
*CELSR1*	94.2	2	0.001	22
*LRP8*	87.8	3	0.029	4
*LCAT*	74.0	4	0.025	8
*SHANK3*	48.3	5	0.026	7
*LAMA5*	–	–	0.041	2
*SP100*	0	11	0.037	3
*LAMA4*	–	–	0.028	5

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
