# Peer review of "Contribution of a Novel B3GLCT Variant to Peters Plus Syndrome Discovered by a Combination of Next-Generation Sequencing and Automated Text Mining"

_ijms, 2019, doi:10.3390/ijms20236006_

Round 1

Reviewer 1 Report

The paper by Toton-Zuranska is well written and the data is well presented. This paper is a case report. The clinical description of the proband is complete and very well presented. The genetic analysis is also reported in full detail. For instance, Table 1 nicely presents the complete data about potential disease-causing variants. Even though there is a single new variant associated with Peters Anomaly described in this paper, given that Peters Anomaly is a very rare eye disorder, the data is worth publishing.

However, this reviewer does not understand the emphasis put on the pipeline to detect the variants. B3GLCT has been previously associated with Peters Anomaly and should therefore be included in any genetic analysis of patients affected by anterior segment dysgenesis. One of the two B3GLCT variants has already been reported to be pathogenic. The newly reported nonsense-variant in the B3GLCT gene is classified as likely pathogenic or pathogenic according to the ACMG 2015 classification, and this could have been simply defined by InterVar or Varsome online tools. The pipeline presented in this paper to prioritize variants identified by high-throughput sequencing is a nice effort, but in the present case established molecular diagnostic tools would have easily concluded that the two identified B3GLCT variants are most likely causal.

This reviewer asks therefore that the variants are also analyzed according to ACMG 2015 classification in molecular diagnosis setting, and the Abstract and Results to be adjusted accordingly. The discussion also could then be shortened.

Author Response

We would like to thank this reviewer for the insightful comments. We have provided our response below and modified the text of the manuscript accordingly. We agree that using publicly available tools is not very innovative ,however used in combination with literature mining software like AMELIE may help researchers and clinicians in the interpretation of next generation sequencing data, which is especially complicated in the case of patients with multisystemic manifestation of genetic changes. First it prioritizes variants according to the current knowledge from published data and second it enables critical verification of results through linking ranked genes important for phenotype with relevant publication, what accelerate the process and may reduce false positive findings. We provided information about compliance with ACMG recommendations in the text on pages 2, 5, and 6. We removed several sentences with redundant information from the discussion section.

Reviewer 2 Report

Authors report single case of patient with anterior segment dysgenesis. They performed next generation sequencing in proband and both parents. Proband was found to be compound heterozygote for one already known and one novel mutation in B3GLCT gene.

However, this study did not bring novel knowledge except for one novel mutation. B3GLCT gene is already known and was originally discovered in 2006 by Lesnik Oberstein et al.

There are several panels and filtering strategies for NGS data evaluation, like integration of information from databases like OMIM, ClinVar etc. and the flow chart described by authors is not very innovative.

Author Response

We would like to thank the reviewer for this opinion. Except of information on novel variant in B3GLCT gene being the cause of Peters Plus Syndrome in our patient, the study presents possible analytical scheme, which could be used by others to improve and accelerate searching for the causal variant. We agree that external databases like OMIM and ClinVar contain information on genes and variants, however the information on variants pathogenicity attributions is not always accurate and on the other hand databases which are manually curated enrich the existing information with new findings carefully at moderate pace. In addition, combination of these information provides an opportunity for quantitative prioritization of the variants. Thus we suggest that critical searching of the literature could make the concluding more accurate and it is facilitated by AMELIE, we used in our study.

Reviewer 3 Report

The paper by Totoń-Żurańska, J. is interesting. In my opinion, what is till required to improve the manuscript:

a proper decription of the softwares they have used. A detailed description of the utilities of the software will enable the researchers in the same or other field to use these softwares for the future studies. Image quality should be improved. I can not see any scale bars either in the Fig.2 or Fig. 3. Why the Figure 2 starts with the number 2, I donot see any Figure before that one. I think the result and discussion sections should have a better explanation of the findings. Introduction section is also very short. I do not see any description of the importance of the genes they have found out in their paper.

Author Response

We would like to thank this reviewer for insightful comments. We made every effort to improve our manuscript according to these suggestions. According to Reviewer’s suggestions scale bars were added and figures’ numeration was  corrected. Figure 5 contains a clear description of the steps in our pipeline and the software that we used. Additionally methods section contain the detailed information on the tools including software version. Also introduction section was improved to better explain the existing state of knowledge. We also add the information on the variant pathogenicity according to ACMG 2015 to show the importance of discovered variants on PPS in the discussion section.  

Round 2

Reviewer 2 Report

This study did not bring novel knowledge except for one novel mutation. B3GLCT gene is already known and was originally discovered in 2006 by Lesnik Oberstein et al. and did not improve significantly after review.

There are several panels and filtering strategies for NGS data evaluation, like integration of information from databases like OMIM, ClinVar etc. and the flow chart described by authors is not very innovative.